# Vegetation Growth Trends of Grasslands and Impact Factors in the Three Rivers Headwater Region

**Xiaoping Sun** [1] **and Yang Xiao** [2,*]

1 Jiangsu Key Laboratory for Bioresources of Saline Soils, Jiangsu Synthetic Innovation Center for Coastal Bio-Agriculture, Jiangsu Provincial Key Laboratory of Coastal Wetland Bioresources and Environmental Protection, School of Wetland, Yancheng Teachers University, Yancheng 224001, China

2 China-Croatia "Belt and Road" Joint Laboratory on Biodiversity and Ecosystem Services, CAS Key Laboratory of Mountain Ecological Restoration and Bioresource Utilization & Ecological Restoration and Biodiversity Conservation Key Laboratory of Sichuan Province, Chengdu Institute of Biology, Chinese Academy of Sciences, Chengdu 610041, China

\* Correspondence: xiaoyang@cib.ac.cn

**Abstract:** Areas of grassland improvement and degradation were mapped and assessed to identify the driving forces of change in vegetation cover in the Three Rivers headwater region of Qinghai, China. Based on linear regression at the pixel level, we analyzed the vegetation dynamics of the grasslands of this region using MODIS NDVI data sets from 2000 to 2010. Correlation coefficients were computed to quantitatively characterize the long-term interrelationship between vegetation NDVI and precipitation/temperature variability during this period. The use of time series residuals of the NDVI/precipitation linear regression to normalize the effect of precipitation on vegetation productivity and to identify long-term degradation was extended to the local scale. Results showed that significant improvements occurred in 26.4% of the grassland area in the Three Rivers Headwater region between 2000 and 2010. The study area, which represents about 86.4% of the total grassland area of this headwater region, showed a general trend of improvement with no obvious trend of degradation.

**Keywords:** NDVI; grassland; MODIS; precipitation variability; human activities

## 1. Introduction

The impact of climate change is multi-scale, all-round and multi-level, with both positive and negative impacts, but its negative impacts are more concerned. After entering the 21st century, climate change has gradually become a comprehensive problem affecting the whole world. As a result of climate change, some grasslands around the world are already experiencing a decline in primary productivity and biodiversity, which is caused by man-made activities and also caused by the climate. These two causes interact and produce a superimposed effect. On the one hand, due to the dry climate, which affects the adaptability of the grassland ecosystem itself, human activities have largely changed the native grassland, which also leads to the decline of the adaptability of the grassland ecosystem itself. A large number of new varieties introduced by artificial and semi-artificial pastures need strict management and protection to create higher pasture yield. Under the climate change model, frequent pasture management measures exert great pressure on the grassland ecosystem. On the other hand, climate change makes the grassland ecosystem itself face more natural disturbances. Climate change affects precipitation and grassland temperature, which causes instability such as year-round drought, summer floods, and winter snowfall. For instance, Orusa and Mondino studied climate change effects on rangelands, and found that phenological and evapotranspiration-related processes and snowpack melting time have been dramatically changed in the last two decades in Aosta Valley (Orusa and Mondino, 2021). Located in the hinterland of the Qinghai-Tibet Plateau

in southern Qinghai, the source area of the three rivers refers to the source areas of the Yangtze, Yellow, and Lancang Rivers [1]. This headwater area is the highest, largest, and most concentrated wetland in the world and has the most abundant biodiversity and the most sensitive ecosystem in China. The three rivers supply about 60 billion m$^3$ of water each year, and the partial water of the Yangtze River, Yellow River, and Lancang River comes from the Three Rivers Headwater Region, thus establishing the area as a veritable "Chinese water tower". The Three Rivers headwater region is a unique area that affects the development of the west; once damaged, recovery is difficult because of the harsh conditions found there and the area's fragile ecosystems [2].

Grassland is the dominant ecosystem in the Three Rivers headwater region, and grassland animal husbandry is the leading industry. One of the main supply functions of this grassland ecosystem is providing herbage allowance for animal husbandry production, which then provides direct benefits to the people through the production of grass-fed livestock [3,4]. In recent decades, due to the impact of the uncontrolled exploitation and overuse by humans, serious grassland degradation has occurred [5]. By the early part of this century, degraded grassland occupied 26–46% of available grassland, and this has had severe impacts on the ecological environment, its security, and on the sustainable development of grassland animal husbandry of the region. To address this problem, the State Council in 2005 approved the Ecosystem Conservation project in the Three Rivers headwater region with the aim of implementing ecological restoration [6]. To design a scientific strategy for the recovery, management, and utilization of the grassland and to effectively evaluate the project [7], analyzing the dynamic change of grassland productivity before and after the start of regional restoration work is essential. This allows an assessment of the trends of the supply function of herbage allowance and the natural and cultural drivers leading to the changes of the grassland ecosystem [8].

A vegetation index refers to ground vegetation coverage gathered through satellite remote sensing; it is a comprehensive, abstract, and indirect method [9] that is accomplished either through empirical modeling or mixed-pixel decomposition [10,11]. An empirical model is applied to achieve vegetation coverage over a large area through the correspondence between the measured data of vegetation coverage in the sample and the vegetation index [12]; however, the application of this method is subject to temporal conditions. With mixed-pixel decomposition, pixel information gathered by remote sensing is simplified into information or non-information on vegetation, and vegetation coverage is estimated from the proportion of vegetation information [13,14]. Studies have shown that this method is not subject to the latest data, and so it is generally applied to dynamic monitoring of remote sensing of vegetation coverage [15].

In this paper, we integrate various remote sensing analytical techniques (trend analysis, correlation analysis, and residual analysis with consideration of hysteresis effect of rainfall on vegetation) to map and assess grassland improvement and degradation areas to identify the driving forces of change in vegetation cover in the Three Rivers headwater region.

## 2. Study Area

The Three Rivers headwater region (TRHR) refers to the source areas of the Yangtze River, the Yellow River, and the Lancang River [16,17]. The TRHR is the highest, the largest, and the most concentrated wetland in the world, while also being the region with the most abundant biodiversity and the most sensitive ecosystem owing to its varied topography, complex hydrographic network, and numerous lakes. Wetland and meadow plants are the main types of vegetation in the TRHR, and these play a vital role in water conservation, runoff mitigation, and biodiversity maintenance [18].

Additionally, as the primary water source of the Yangtze, Yellow, and Lancang rivers, the Three Rivers headwater region, named "China water tower", is vital for maintaining the water security of 548 million people on the downstream Yellow River basin, Yangtze River basin and Lancang River basin. However, since 1990s, the ecosystem of the Three Rivers headwater region has shown a severe trend of degradation, which seriously threatens

the production and living of the local residents in the downstream areas. Therefore, the Chinese government has started to implement a series of ecological policies to curb the degradation of the ecosystem (Figure 1).

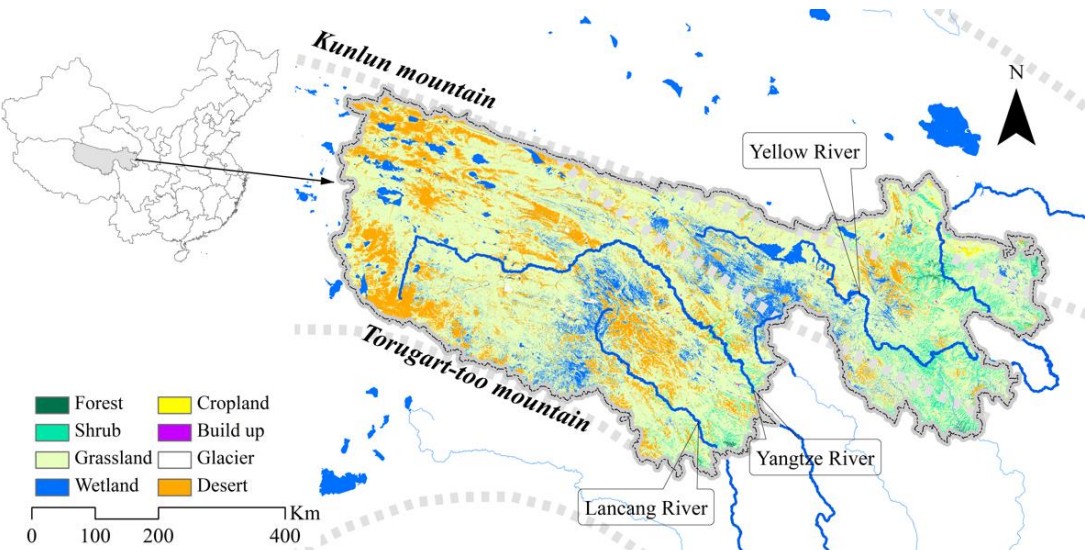

**Figure 1.** Grassland distribution of the Three Rivers headwater region (The gray bold dashed lines represent the mountains with a name beside it; the blue lines are the rivers; spatial reference: Albers conical equal area; datum: D WGS 1984).

## 3. Materials and Methods

### 3.1. Data Sets and Pre-Processing

MODIS data: The normalized difference vegetation index (NDVI) is a normalized ratio of red and near-infrared reflectance. It is used as a terrestrial vegetation growth and conditions proxy, being sensitive to chlorophyll-related plant canopy structural variations and being closely correlated to the fraction of potential photosynthesis and the physiological status of vegetation [19,20]. In this study, we used the moderate resolution imaging spectroradiometer (MODIS) MOD13Q1 time series data set from 2000 to 2010 obtained from the Land Processes Distributed Active Archive Center (LP DAAC). This data set has a spatial resolution of 250 m and temporal resolution of 16 days, an improved sensitivity to vegetation, and a reduced influence of external factors (such as atmosphere, observation angle, sun azimuth, and cloud cover), which have been verified using stable desert control points.

The MODIS data set was geocoded to the universal transverse Mercator (UTM) coordinate system using the MODIS reprojection tool. As these products may be affected by cloud cover, atmosphere, or ice/snow cover, we first reconstructed the NDVI time series data set using the asymmetric Gaussian function filter in TIMESAT 2.3 software program to reduce noise and improve data quality during the data pre-processing procedure [21]. The 16-day MODIS-NDVI was then compiled into monthly NDVI data by applying maximum value compositing (by overlaying multiple raster maps, the value of each raster cell is then taken as the largest of the multiple maps), which was processed in the interactive data language (IDL, https://www.harrisgeospatial.com, 30 November 2022).

Meteorological data: Meteorological data from 2000 to 2010 (including monthly scale precipitation and temperature) were obtained from the Chinese National Metrological Information Center/China Meteorological Administration. Monthly meteorological data derived from station-based information were interpolated to the whole research area at a spatial resolution of 250 m using the kriging interpolation method. In addition, the monthly temperature was calibrated using the digital elevation model, and the coefficient

was reduced by a 0.47 degree/100 m increase in elevation. This coefficient was obtained by linear regression between elevation and temperature.

Grassland data: Information related to grassland distribution in the study area was obtained from an ecosystem map of China that was generated and interpreted based on 1:100,000 Landsat TM satellite remote sensing products from the Data Center for Resources and Environmental Sciences, Chinese Academy of Sciences. (Figure 1). Sources of cartographic data and statistics are listed in Table 1.

**Table 1.** Sources of principal data.

| Data Name | Data Resolution | Data Source |
|---|---|---|
| MODIS-NDVI | 250 m (monthly) | Land Processes Distributed Active Archive Center (https://lpdaac.usgs.gov, 30 November 2022) |
| SRTM | 90 m | CGIAR Consortium for Spatial Information (http://srtm.csi.cgiar.org/, 30 November 2022) |
| Precipitation, temperature | 146 points (monthly) | Chinese National Metrological Information Center/China Meteorological Administration (http://data.cma.cn, 30 November 2022) |
| Ecosystem map | 90 m (yearly) | Resource and Environment Science and Data Center, Chinese Academy of Sciences (https://www.resdc.cn/, 30 November 2022) |

*3.2. Methods*

(1) Trend analysis

To assess variation trends of NDVI and climate (precipitation and temperature) throughout the 2000–2010 study period, we used a linear least-squares regression model to obtain the changing trends of every pixel by fitting a linear equation of NDVI or climate variables as a function of the variable of year to obtain an image of changing slopes [22]. The linear least-squares regression method, which is a commonly used method in trend analysis [23], was applied as follows:

$$y = a + b \times t + \varepsilon \tag{1}$$

where $y$ represents NDVI or climate variables; $t$ is year; $a$ and $b$ are fitted variables ($b$ is the slope as a proxy of trend and a is the intercept); and $\varepsilon$ is the residual error. If $b > 0$, there is an increasing trend of NDVI or climate; conversely, if $b < 0$, there is a decreasing trend. $p < 0.05$ was considered a significant change for both increasing and decreasing trends (Table 2).

**Table 2.** Evaluation standard of trend significance.

| Variation Trend | $b$ Value Range | $p$ Value Range |
|---|---|---|
| Significant decrease | $b < 0$ | $p \leq 0.05$ |
| Significant increase | $b > 0$ | $p \leq 0.05$ |
| No significant change | | $p > 0.05$ |

In addition, the estimation of parameters $a$ and $b$ uses the least square method, $\varepsilon_i$ is a random error, the fitting value of parameters $a$ and $b$ is expressed as:

$$\hat{b} = \frac{\sum_{i=1}^{n} (t_i - \bar{t})(y_i - \bar{y})}{\sum_{i=1}^{n} (t_i - \bar{t})^2} \tag{2}$$

$$\hat{a} = \bar{y} - \hat{b} \times \bar{t} \tag{3}$$

The calculation process of changing slopes per pixel was programmed with interactive data language (IDL).

(2)    Correlation analysis

We used the Pearson correlation coefficient to explore the relationship between trends of NDVI and climate as follows:

$$r = \frac{\sum_{i=1}^{n}(x_i - \overline{x})(y_i - \overline{y})}{\sqrt{\sum_{i=1}^{n}(x_i - \overline{x})^2}\sqrt{\sum_{i=1}^{n}(y_i - \overline{y})^2}} \tag{4}$$

where $r$ represents the linear correlation coefficient; $x$ is the NDVI variable; $y$ is the precipitation or temperature variable; and $n$ is the number of variables [24].

To evaluate the influence of anthropogenic activities on the trend in NDVI, we used partial correlation between the slope of the NDVI and anthropogenic factors (i.e., population growth and a change in livestock numbers). We also employed stepwise regression between the trend in NDVI and the impact factors (precipitation, temperature, population, and livestock numbers) to assess their relative contribution in influencing spatial characteristics of the NDVI trend [25].

(3)    Residual analysis

Residual NDVI is the difference between the observed value $y_i$ (observed NDVI) of the dependent variable and the predicted value $\hat{y}_i$ (predicted NDVI) calculated according to the estimated regression equation, denoted by $e$. It reflects the error caused by using the estimated regression equation to predict the dependent variable $y_i$. The residual of the $i$th observation is:

$$e_i = y_i - \hat{y}_i \tag{5}$$

Residual analysis is used to investigate trends in residual differences (residual NDVI) using a regression model involving. The trends of predicted NDVI were interpreted as climate-induced changes (rainfall as the explanatory variable [26]), while trends in residual NDVI ($e_i$) were interpreted as human-induced changes [26].

The residual analysis mainly involved the following steps. First of all, a line regression model between observed NDVI and precipitation factor was used to calculate the residual difference ($e$) between observed NDVI and predicted NDVI (regression result). Then, the trend analysis of residuals as a function of time was processed to investigate the human-induced vegetation degradation (Table 3). Pixels exhibiting marginal decreases (i.e., <5%) in 2000–2010 were excluded or considered as stable because they reflected the potential uncertainties caused by differences due to image dates within the season/month or image calibration processes.

**Table 3.** Indicators and definitions related to residual analysis methods.

| Degradation Trends of Observed NDVI | Trends of Predicted NDVI | Slope of Residual | Definition Description |
|---|---|---|---|
| <0 | >0 | <0 | Human-induced vegetation degradation |
|  | <0 | >0 | Climate-induced vegetation degradation |
|  | <0 | <0 | Both climate- and human-induced vegetation degradation |
|  | >0 | >0 | Uncertainty error |

Previous studies reported that annual maximum NDVI representing the growth of grasslands is strongly correlated with climatic variables [26]. Thus, $NDVI_{max}$ as the highest NDVI value can be used to gauge grassland vegetation growth in this study. Based the residual analysis, measured $NDVI_{max}$ values showed both positive and negative deviations from the fitting curve on the $NDVI_{max}$/rainfall linear regression, suggesting that vegetation is not only responsive to rainfall, but also influenced by human activities represented by residuals.

As the rainfall period was most strongly correlated with grassland growth, we calculated the precipitation accumulation periods and lag periods from September to August

of next year. And analysis the correlation between the cumulative rainfall and NDVI$_{max}$, to identify the optimal relationship between them in the study area [27]. By prolonging and changing the cumulative period, the process was repeated until all possible combinations were tested, and then the best relevant cumulative period was determined (with the strongest correlation), which is from 1 September to 1 August of next year.

## 4. Results

### 4.1. Spatial and Temporal Characteristics of Grassland Variation

Trend analysis showed overall positive NDVI trends for the grasslands (Figure 2). The study area, which is about 84.25% area of the total grassland in the Three Rivers headwater region, showed a continuous trend of improvement from 2000 to 2010.

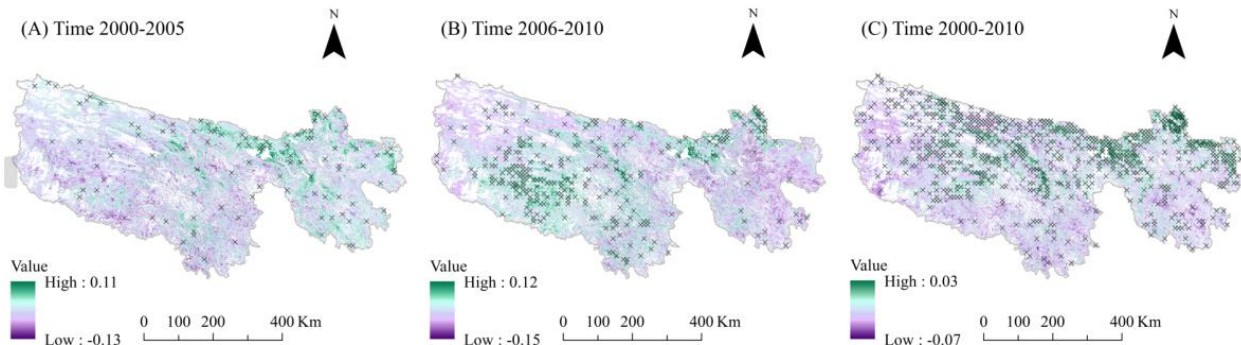

**Figure 2.** Spatial distribution of NDVI trends in the grassland ecosystem. (**A**) NDVI trend from 2000 to 2005; (**B**) NDVI trend from 2006 to 2010; (**C**) NDVI trend from 2000 to 2010; the fork symbol in the figure indicates the significance of NDVI trend, where *p* value < 0.05. (Spatial reference: Albers conical equal area; datum: D WGS 1984).

Linear regression analysis of NDVI from 2000 to 2005 indicated that about 37.01% of the total grassland area experienced a declining trend in vegetation production (Figure 2A). The declining patches of vegetation were found mainly in the western and southern pastures; however, this change between 2001 and 2005 was not significant. On the contrary, the long-term productivity of vegetation shows a positive or stable trend in the northern high-altitude pastoral areas affected by fog and haze.

Correlation between grassland degradation and triggered factors:

The correlation between NDVI values and accumulated precipitation revealed a strong positive correlation (r = 0.5 to 0.98) for about 38.41% of the grassland area in the Three Rivers headwater region (Figure 3A). A negative correlation was found in the densely populated southern and eastern parts of the study area, which have low vegetation cover and high levels of human activity.

Moreover, there is little correlation between residual NDVI and rainfall, suggesting that residual analysis can effectively remove climatic factors (precipitation) and effectively reflect the impact of human activities on grassland degradation (Figure 3B).

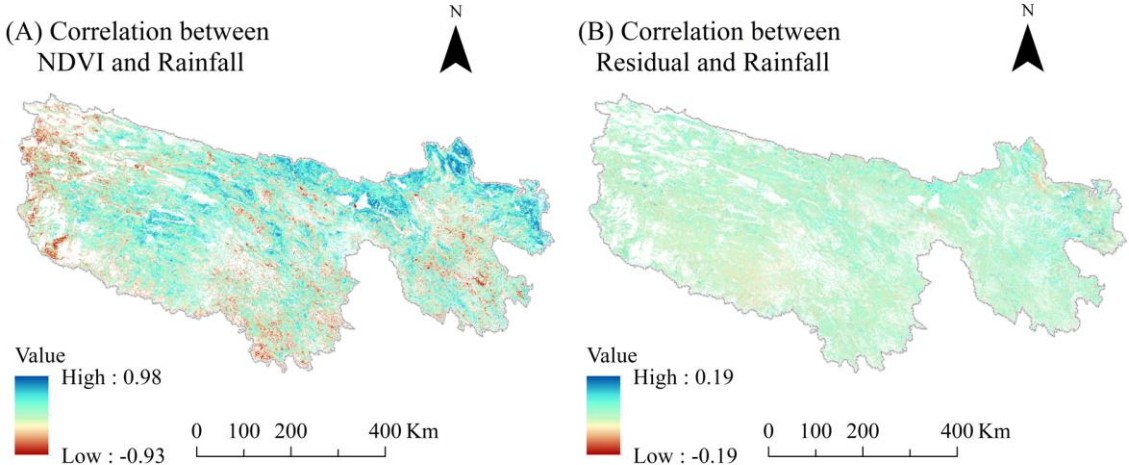

**Figure 3.** Correlation results based on person correlation analysis. (**A**) Person correlation between NDVI and Rainfall; (**B**) Person correlation between residual NDVI and Rainfall. (spatial reference: Albers conical equal area; datum: D WGS 1984).

*4.2. Impacts of Human Activities on Grassland Degradation*

Analysis showed that approximately 25.85% of the study area did not show an increase in vegetation growth despite increasing precipitation from 2000 to 2010 (Table 4). The declining patches of vegetation, which can be attributed to human activities, were found mainly in the southern and eastern pastures (Figure 4A). When evaluating the first and second halves of the decade, linear regression analysis of residual NDVI from 2000 to 2005 showed that about 35.72% of the total grassland area experienced a declining trend in vegetation production (Figure 4B). In contrast, about 17.66% of the total grassland area showed a declining trend in vegetation production from 2006 to 2010 (Figure 4C).

**Table 4.** Proportion of area showing non-climate degradation (units %) and the trend of the residual mean value.

|  | 2000–2010 | 2000–2005 | 2006–2010 |
|---|---|---|---|
| Non-climate degradation area (%) | 25.85 | 35.72 | 17.66 |
| Mean of residual slope | −27.35 | −88.51 | −62.10 |

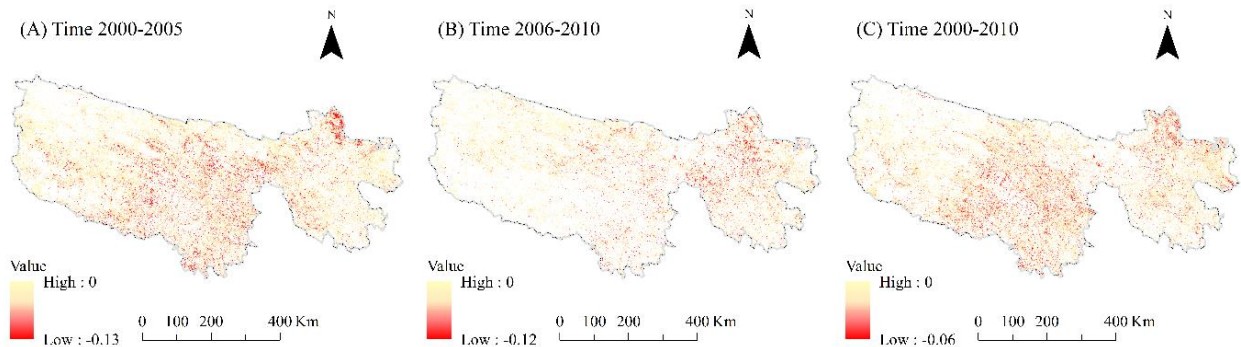

**Figure 4.** Grassland degradation trend due to human activities (**A**) residual NDVI from 2000 to 2005; (**B**) residual NDVI from 2006 to 2010; (**C**) residual NDVI from 2000 to 2010. (spatial reference: Albers conical equal area; datum: D WGS 1984).

**5. Discussion**

Our findings showed that an increasing NDVI occurred mainly in the northern plains, while southern and eastern grazing pastures, which are densely populated, showed a small positive trend in long-term vegetation productivity. Degraded patches of vegetation were

located on steep slopes (slope > 20°), supporting the findings of Fan et al. (2010) [16]. In addition, our field observations suggested that high wind erosion and intensive human activities were impacting vegetation productivity. Grassland is the most important ecosystem type on the Qinghai Tibet Plateau and the basis of animal husbandry on the plateau. Though a series of grazing grassland ecological protection subsidy incentives such as grassland ecological protection in the construction project, the Qinghai Tibet Plateau grassland conservation effect appeared gradually, but the strength of the human activity factor is different, we also found that declining patches of vegetation were found mainly in the pastures [28].

A large proportion of the total grassland area (80.79%) showed an increasing vegetation trend between 2000 and 2005. Less than 20% of the total grassland showed degradation, and this was patchy rather than continuous. By 2006, the extent of vegetation coverage was even higher. We attribute this improvement to the 2005 approval of the Ecosystem Conservation project in the Three Rivers headwater region [29], which has led to less human disturbance and a focus on vegetation recovery [6,30]. Whether changes in precipitation variability or human activities led to this recovery is not yet clear. However, we found only a limited correlation between precipitation and residual NDVI (Figure 4b), indicating that the trends in vegetation cover were unlikely to be caused as a result of climatic conditions.

To analyze temporal trends in grassland NDVI, we concentrated on the development of the annual maximum NDVI, as proposed by Evans and Geerken (2004) [25]. Vegetation production in a cold alpine environment has been shown to fluctuate strongly according to interannual precipitation and temperature variability [16,31]. However, temperature is periodic and takes years for little change. Therefore, we assume that the correlation between temperature and vegetation growth is unlikely to be used to identify the temporal and spatial trends of grassland, so we ignore temperature. The NDVImax values are calculated and correlated with precipitation, as proposed by Evans and Geerken (2004) and Madonsela et al. (2018) [26,32]. We found that a decreasing NDVI was mainly concentrated in the western region due to enhancing human activity such as over grazing and an increasing NDVI was distributed throughout the rest of the study area due to increasing rainfall and ecological protection policies such as banning grazing [33].

Our findings document the benefits that have occurred since the initiation in 2006 of the ecological restoration project, which included returning grazing land to grassland, the ecological migration project, black soil land recovery (grassland with extreme degradation), and rodent control [16,34]. Non-climatic (human) activities led to serious grassland degradation of the Three Rivers headwater region up until 2005 (Figure 4B), but since 2006, the grassland degradation caused by these activities has decreased and an increase in vegetation cover has occurred. This finding combined with the decreasing residual NDVI indicates that the increasing vegetation cover over time is not related to precipitation variability but to better land management practices.

## 6. Conclusions

We found an overall positive NDVI trend between 2000 and 2010 for the grasslands in our study area, which cover about 84.25% of the total grassland area in the Three Rivers headwater region. The declining trend in vegetation production between 2000 and 2005, which affected about 37.01% of the total grassland area, appears to have been reversed between 2006 and 2010, most likely by the effectiveness of the government-approved program for ecosystem conservation.

The amount of precipitation in the Three Rivers headwater region has a strong positive correlation with NDVI in about 38.41% of the grassland area. Negative correlations were found in the southern and eastern regions, which have dense populations, low vegetation coverage, and intensive human activity. Our study shows that to promote grassland recovery in Three Rivers headwater region, a number of factors must be taken into account, especially human activities. Our findings also support the need for intensive management

of water resources to enhance vegetation and avoid ecological damage in this important ecological area.

**Author Contributions:** Y.X. conceived and designed the research; X.S. and Y.X. analyzed the data; X.S. and Y.X. wrote the main manuscript; X.S. prepared Figures and Tables. All authors reviewed the manuscript. X.S. conceived and designed the experiments, performed the experiments, contributed reagents/materials/analysis tools, analyzed the data, prepared figures and/or tables, and wrote the main manuscript. Y.X. conceived and designed the experiments, contributed reagents/materials/ analysis tools, wrote the main manuscript, authored or reviewed drafts of the paper, and approved the final draft. All authors have read and agreed to the published version of the manuscript.

**Funding:** This work was funded jointly by the National Natural Science Foundation of China (NSFC) (No: 71904060), the General Project of Natural Science Research of Jiangsu Province Higher Education Institutions (22KJD180005), the Natural Science Foundation of Jiangsu Province (BK20211363), opening grant of Jiangsu Key Laboratory for Bioresources of Saline Soils (JKLBZ202001).

**Data Availability Statement:** The raw/processed data required to reproduce these findings cannot be shared at this time as the data also form part of an ongoing study.

**Conflicts of Interest:** The authors declare no competing interest.

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
