# Peer review of "Vegetation Growth Trends of Grasslands and Impact Factors in the Three Rivers Headwater Region"

_land, doi:10.3390/land11122201_

Round 1

Reviewer 1 Report

The manuscript is well written and well structured minor changes have to be performed before going ahead in the pubblication phase.

Firstly, I suggest you to better discuss climate change effects on grassland worldwide in the introduction. Therefore, I deeply suggest you to consider to include this manuscript into the introduction section. 

What specific improvements should the authors consider regarding the methodology? What further controls should be considered?

Are the conclusions consistent with the evidence and arguments presented and do they address the main question posed?

- https://doi.org/10.3390/cli9030047

https://doi.org/10.1007/978-3-642-57030-8_5

Then, please provide for each maps the reference system and the datum as well as the nominal and representation scale. You may include them in the caption section below or into the maps.

Finally, I suggest you to create a separate from material and methods section in which you report the area of study. 

Reviewer 2 Report

The authors tried to explore the relationships between NDVI and climate factors using regression models. This is very common in existing literature. Considering the study area which might be not studied by other researchers, I think it could be fine if some interesting results are reported. My big concern is the regression models because the regression was performed at pixel basis and the period was too short that too few samples were included in the analysis. Details are given in the following:

Regressions with 5 years (2000-2005 and 2005-2010) are not appropriate, because there are only 5 samples.  Based on statistics principle  too few observations can lead to overfitting. 

The regression between NDVI and precipitation (temperature) was performed at pixel basis for 2000-2005, 2000-2010 and 2005-2010. For each pixel, the residual should follow random distribution without trend. However, "Fig. 4 showed declining trend in the residual NDVI". This means the regression models had a bias problem. I don't think the residual was caused by human activities.

Lines 209-211: Moreover, there is little correlation between residual NDVI and Rainfall, suggesting that residual analysis can effectively remove climatic factors (precipitation) and effectively reflect the impact of human activities on grassland degradation (Fig 3b). Does the residual really come from human activities?

Round 2

Reviewer 2 Report

I thank the authors for the revision and now it looks good to me so I suggest publication. It is still unclear how MOD13Q1 was converted into monthly scale data and readers may be interested in knowing the detail of the process.

Author Response

Responses to Reviewers

Dear editor:

We really appreciate reviewers and you for the really thorough review of our manuscript and totally agree on the detailed comments you mentioned. We have completed the revision according to the review comments. Indeed these modifications have greatly improved the manuscript. Below we will explain point by point how we have addressed the suggestions.

We look forward to your response.

Kind regards, on behalf of the coauthors.

#1 Reviewer

Q1: I thank the authors for the revision and now it looks good to me so I suggest publication. It is still unclear how MOD13Q1 was converted into monthly scale data and readers may be interested in knowing the detail of the process.

Answer: Thank you for your suggestion. Relevant content has been revised in Materials and Methods part:"

The 16-day MODIS-NDVI was then compiled into monthly NDVI data by applying maximum value compositing (By overlaying multiple raster maps, the value of each raster cell is then taken as the largest of the multiple maps) , which was processed in Interactive Data Language (IDL, https://www.harrisgeospatial.com).

In addition, we have further corrected some minor errors in the manuscript, such as the format of the mathematics, misspelling of technical terms, etc., Detailed modification information can be found in the revised manuscript with changes marked.
